# Investigation of Angle Measurement Based on Direct Third Harmonic Generation in Centrosymmetric Crystals

**Kuangyi Li** [1], **Jiahui Lin** [1], **Zhiyang Zhang** [1], **Ryo Sato** [1], **Hiroki Shimizu** [2], **Hiraku Matsukuma** [1,*]
**and Wei Gao** [1]

1    Precision Nanometrology Laboratory, Department of Finemechanics, Tohoku University,
     Sendai 980-8579, Japan
2    Department of Engineering, Graduate School of Engineering, Kyushu Institute of Technology,
     1-1 Sensui-cho, Tobata-ku, Kitakyushu-shi 804-8550, Japan
*    Correspondence: hiraku.matsukuma.d3@tohoku.ac.jp; Tel.: +81-022-795-6953

**Abstract:** This paper proposed angle measurement methods based on direct third harmonic generation (THG) in centrosymmetric crystals. The principles of the intensity-dependent and the wavelength-dependent angle measurement methods were illustrated. In this study, three prospective centrosymmetric crystals and two different phase-matching types were investigated in a wavelength range from 900 nm to 2500 nm. For the intensity-dependent method, a dispersion-less wavelength range was found from 1700 nm to 2000 nm for $\alpha$-BBO and calcite. Compared with rutile, $\alpha$-BBO and calcite had relatively better measurement performance based on the angle measurement sensitivity calculation. The wavelength-dependent method was considered in a dispersive range of around 1560 nm. The results suggested that $\alpha$-BBO and calcite were also suitable for wavelength-dependent measurement. In addition, the effects of focusing parameters were considered in the simulation, and the optimized focal length ($f$ = 100 mm) and the focused position (in the center of the crystal) were determined.

**Keywords:** angle measurement; direct third harmonic generation; centrosymmetric crystals; femtosecond laser

## 1. Introduction

Angle measurement is essential for controlling the quality of products in manufacturing processes [1]. Among angle measurement methods, optical angle measurement is widely adopted for the advantage of non-contact detection. In the decades, an amount of optical angle measurement techniques were developed [2], such as the well-used optical encoders mounted on the rotary shaft for covering the 360° angle displacement range of rotation [3]. Accurate evaluation of small angular displacements is also required in many situations, such as the angular error motion of a moving stage and the tilt error motion of a rotating spindle [4]. For this purpose, angle sensors based on autocollimation and light interferometry were proposed [5–7]. In such an angle sensor, a reflective mirror is mounted on the measured target. The angular displacement of the reflective mirror induces changes in the sensor output for the measurement.

In the field of optical metrology, an appropriate light source should be selected. Femtosecond laser sources have not only the characteristics of conventional laser sources but also a series of stable, equally spaced modes in the frequency domain and femtosecond-scale ultrashort pulses in the time domain [1]. Several angle measurement setups have been built employing the femtosecond laser source for high-resolution angle measurement [8–10]. For the optical lever sensor based on the femtosecond laser, a grating reflector was mounted on a rotary stage while the detector was kept stationary. In this case, the femtosecond laser beam was projected onto the grating reflector after the laser beam was modulated by using a Fabry–Pérot etalon. A group of first-order diffracted beams of the

different modes was then generated at the grating reflector [8]. As the stage was rotated, the detected intensity would change periodically, which extends the measurement range of angle measurement to a maximum of 15,000 arcseconds. Similarly, the mode-locked femtosecond laser autocollimator also extends the angle measurement range by taking the advantage of the widely spreading spectrum of the femtosecond laser source in the frequency domain [9,10]. Compared with the optical lever sensor, the angle measurement sensitivity of the autocollimator is not affected by the distance between the detector and the grating reflector, where a compact setup design can be realized.

Meanwhile, an extremely high electrical field of femtosecond laser induces the nonlinear polarization field which is the source of nonlinear optical (NLO) phenomena, such as the second harmonic generation (SHG) [11]. SHG is a common NLO process in which the second harmonic of an incident laser is generated by passing some kind of material through it. The produced second harmonic wave (SHW) has the double frequency $\nu_2$ ($\nu_2 = 2\nu_1$) and the half wavelength $\lambda_2$ ($\lambda_2 = \lambda_1/2$) of the incident laser beam (fundamental wave), which has the frequency and the wavelength of $\nu_1$ and $\lambda_1$. The angle dependence of SHG in the birefringent crystal was found in the early 1960s [11]. As the laser propagates through the birefringent crystal in different directions, the phase mismatch between the laser and the second harmonic wave changes due to anisotropy, resulting in the intensity changes of the generated SHW. Based on this principle, several prototypes of the SHG-based angle measurement were developed with the commercial mode-locked femtosecond laser [12–14]. In the early sensors, a lens was employed to focus the laser beam into a crystal for reducing the diameter of the light spot for SHW production. At first, Beta Barium Borate ($\beta$-BBO) crystals with low wavelength dispersion of the SHG output were used and a sub-arcsecond resolution was realized [12]. However, the measurement range of the sensor was narrow and the chromatic aberration in the crystal was unavoidable due to the narrow optical pulse width, resulting in a significant reduction in measurement sensitivity. Then, the wavelength-dependent SHG angle sensor was proposed [13], where a shift of the SHG spectrum in the frequency domain was used to denote the angle position. However, the focused laser beam should be very carefully aligned with the NLO crystal, making it difficult to measure a moving target [15]. Then, to deal with this problem, an SHG angle sensor with a collimated femtosecond laser beam was constructed where a second harmonic wave was successfully generated with the collimated laser beam [14].

SHG is related to the symmetry of the crystal, described by the point group of the crystal. The different point groups usually have different forms of second-order optical nonlinear susceptibility $\chi^{(2)}$ for generating a second harmonic wave. Theoretically, the SHG signal can only be detected in a crystal without inversion symmetry, called a non-centrosymmetric crystal [16–18]. Several birefringent crystals have an inversion center, also referred to as centrosymmetric structures. The second-order optical susceptibility $\chi^{(2)}$ should be equal to zero in these crystals [19,20]. Therefore, the SHG process is forbidden in centrosymmetric crystals because SHG is a typical second-order nonlinear optical phenomenon relying on the second-order optical susceptibility $\chi^{(2)}$. Compared with SHG, the direct third-harmonic generation (THG) process, where the laser beam directly generates the third harmonic wave, can be found in any material because the third-order optical susceptibility $\chi^{(3)}$ does not vanish regardless of the symmetry of the material even in the air [21,22]. It has also been proved that THG has a significant angle dependence in both the non-centrosymmetric crystals such as $\beta$-BBO [23] and the centrosymmetric crystals such as fused silica and calcite [24,25]. The conversion efficiency of THG reaches 2.5% when the femtosecond laser has an intensity of $10^{11}$ W/cm$^2$, though it is a weaker process than SHG [26]. Since the angular dependence of THG is generally sensitive near the phase-matching angle [23–26], it is expected to be applied to angle sensors.

In this research, we propose two angle measurement methods based on THG, including the intensity-dependent method and the wavelength-dependent method. This paper presents the proposed methods with centrosymmetric birefringent crystals. The non-centrosymmetric crystals are not discussed in this paper because they are seriously affected

by the non-phase-matched second-order $\chi^{(2)}$ optical processes [26]. The simulation results of investigating the angle measurement performance of different centrosymmetric crystals are reported in this paper. Firstly, the angle dependence principle of direct THG in the centrosymmetric crystal is illustrated for an intuitive understanding. Then, two angle measurement methods are introduced, including the intensity-dependent and wavelength-dependent methods. For theoretical investigation, three centrosymmetric crystals are selected, which could achieve a considerate THG conversion efficiency. Their material specifications are also supported by the literature. A wide wavelength range from 900 nm to 2500 nm is considered in the simulation for angle measurement. In this range, the measurement sensitivity is evaluated to choose the crystals that are suitable for angle measurement. In addition, the effects of focusing parameters, including the focal length and the focused position, are discussed. The optimized measurement configurations are found, and the reasons are briefly discussed.

## 2. The Principle of Angle Measurement Based on Direct THG

### 2.1. The Basic Principle of Angle Dependence of Direct THG in Centrosymmetric Crystal

Figure 1a shows a brief schematic of the THG process in centrosymmetric crystal. The fundamental wave (FW) passes through the crystal and produces a third harmonic wave (THW) by three-order susceptibility $\chi^{(3)}$. The produced third harmonic wave has three times the frequency ($\nu_3 = 3\nu_1$) and the wave number ($k_3 = 3k_1$) of the fundament wave. Its wavelength becomes one-third of the FW accordingly ($\lambda_3 = \lambda_1/3$). The angle dependence of THG is mainly related to the birefringence and dispersion of the crystal's refractive index, which makes the light propagate at different speeds in different directions and wavelengths. Therefore, we introduce the refractive index first. For the common uniaxial crystal, Figure 1b gives the refractive index ellipsoid of the uniaxial crystal, where *X-Y-Z* is the Cartesian coordinates of the crystal, and the *Z* axis is assumed to be parallel to the optical axis. $\theta$ is the angle between the wave vector $\boldsymbol{k}$ and the optical axis. $\varphi$ denotes the azimuthal angle, which is the angle between the projection of $\boldsymbol{k}$ in the *X-Y* plane and the *X*-axis. $N_o$ and $N_e$ are the length of the refractive ellipsoid. There are two different propagated waves in crystal, called the ordinary wave (o-wave) and the extraordinary wave (e-wave), of which the electrical fields are perpendicular and parallel to the incident plane, respectively. It should be noted that the effect of the walk-off is ignored in this paper. Thus, we assume that the e-wave is perpendicular to the $\boldsymbol{k}$. The refractive index can be calculated by Equation (1), where the numbers 1 and 3 correspond to the FW and THW. We can see that the refractive index of the e-wave is dependent on the angle $\theta$, and it is also related to the wavelength $\lambda$. Because the refractive index of the e-wave is determined by both $\theta$ and $\lambda$, the angle $\theta$ can be used for the compensation of refractive index dispersion between FW and THW. That is to say, the refractive index $n_3$ of the third harmonic wave may be equal to that of the fundamental wave $n_1$ by propagating at one special angle and choosing a suitable polarization combination, which is so-called birefringent phase-matching.

$$n_{1e,\,3e}(\lambda_{1,3}, \theta) = \left( \frac{\cos^2\theta}{[N_o(\lambda_{1,3})]^2} + \frac{\sin^2\theta}{[N_e(\lambda_{1,3})]^2} \right)^{-1/2} \mathrm{e-wave} \quad n_{1o,\,3o} = N_o(\lambda_{1,3}) \; \mathrm{o-wave} \tag{1}$$

Figure 2 illustrates the type I phase-matching of positive uniaxial crystal as an example. In this figure, FW and THW are assumed as e-wave and o-wave, respectively. *X'-Y'-Z'* and *X-Y-Z* denote the stationary laboratory's coordinate and the rotated crystal coordinate, respectively. The *Z'* axis is parallel to the FW propagation direction $\hat{k}_{1e}$ (laser direction) in crystal. Figure 2a shows the refractive index surface of the phase-matching case. We can see $\hat{k}_{1e}$ is in the direction of phase-matching angle $\theta_m$, which makes the phase mismatching $\Delta k(\theta_m) = 6\pi/\lambda_1(n_{3o} - n_{1e}(\lambda_1, \theta_m)) = 0$ because of the refractive index difference $\Delta n(\theta_m) = n_{3o} - n_{1e}(\lambda_1, \theta_m) = 0$. In the phase-matching case, as given in Figure 2b, FW and THW have the same propagated speed in the crystal because $\Delta k = 0$ and $\Delta n = 0$. The THW generated in different positions (the first, second, and third, positions shown in Figure 2b)

can be strengthened by constructive interference. Therefore, the detectable THG intensity is generated. However, as the crystal is rotated at an angle $\Delta\theta$ away from the phase-matching angle $\theta_{\mathrm{m}}$ shown in Figure 2c, d, the transmitted THG will be weakened because the generated THWs in different positions are reduced by the destructive interference where the $\Delta k(\theta_{\mathrm{m}} + \Delta\theta)$ and $\Delta n(\theta_{\mathrm{m}} + \Delta\theta)$ are not equal to zero. For the birefringent phase matching, there are three types for both the negative uniaxial crystal and the positive uniaxial crystal. Here, we only discuss the type I phase-matching of positive uniaxial crystal and the type II phase-matching of negative uniaxial crystal, taking into consideration that the high-efficiency THG in the centrosymmetric crystals has been demonstrated in the literature. The concept is similar in both the type I phase-matching of positive uniaxial crystal and the type II phase-matching of negative uniaxial crystal. The only difference is that the phase-matching angle $\theta_{\mathrm{m}}$ should meet the condition of $n_{1\mathrm{o}} + n_{1\mathrm{o}} + n_{1\mathrm{e}}(\lambda_1, \theta_{\mathrm{m}}) = n_{3\mathrm{e}}(\lambda_3, \theta_{\mathrm{m}})$, and the incident fundamental wave should contain both the o-wave and the e-wave for the THG process. Combined with Equation (1) and the concept of phase-matching, the calculation of $\theta_{\mathrm{m}}$ can be easily induced, and they are displayed in Equation (2) and Equation (3). Equation (2) is for the type I phase-matching, and Equation (3) is for the type II phase-matching. It should be noted that Equation (3) only shows an implicit relation, which should be solved by numerical methods.

$$\theta_{\mathrm{m}} = \arcsin\left(\sqrt{\frac{[N_{\mathrm{o}}(\lambda_3)]^{-2} - [N_{\mathrm{o}}(\lambda_1)]^{-2}}{[N_{\mathrm{e}}(\lambda_1)]^{-2} - [N_{\mathrm{o}}(\lambda_1)]^{-2}}}\right) \text{ Type I (Positive crystal)} \tag{2}$$

$$\left(\frac{\cos^2\theta_{\mathrm{m}}}{[N_{\mathrm{o}}(\lambda_3)]^2} + \frac{\sin^2\theta_{\mathrm{m}}}{[N_{\mathrm{e}}(\lambda_3)]^2}\right)^{-1/2} = \frac{1}{3}\left\{2N_{\mathrm{o}}(\lambda_1) + \left(\frac{\cos^2\theta_{\mathrm{m}}}{[N_{\mathrm{o}}(\lambda_1)]^2} + \frac{\sin^2\theta_{\mathrm{m}}}{[N_{\mathrm{e}}(\lambda_1)]^2}\right)^{-1/2}\right\} \tag{3}$$

$$\text{Type II (Negative crystal)}$$

Generally, a small incident angle deviation $\Delta\theta$ can cause a large change in the THG intensity, which is especially beneficial for the small-angle measurement. Next, we propose to utilize the angle dependence of direct THG for angle measurement. Two measurement methods will be explained in detail.

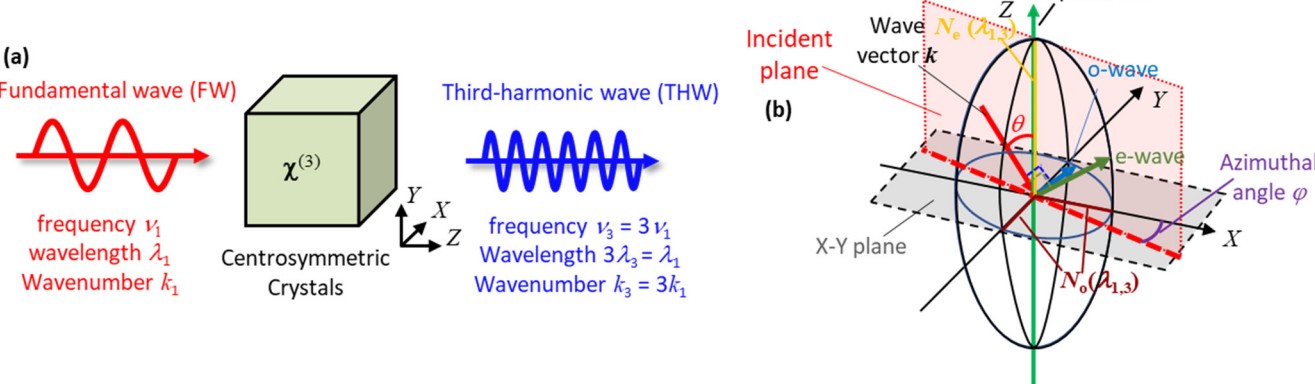

**Figure 1.** (**a**) The schematic of THG by $\chi^{(3)}$ in centrosymmetric crystal; (**b**) The refractive index ellipsoid of uniaxial crystal.

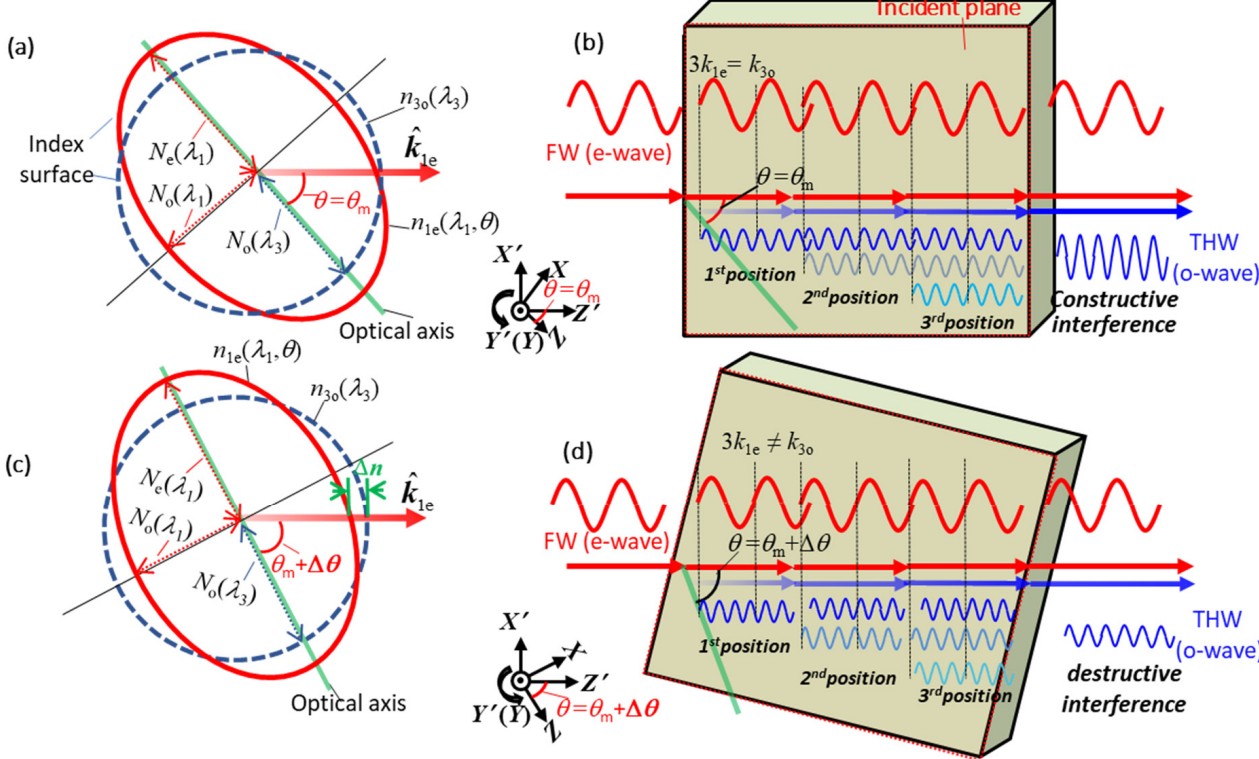

**Figure 2.** (**a**) The refractive index surface of type I phase-matching case; (**b**) The type I phase-matching process in a crystal; (**c**) The refractive index surface of type I phase-mismatching case; (**d**) The type I phase-mismatching process in crystal.

### 2.2. The Angle Measurement Methods Based on Direct THG with a Femtosecond Laser

Figure 3a shows the angle measurement schematic based on direct THG in centrosymmetric crystal. Because the magnitude of $\chi^{(3)}$ for THG is very small, the femtosecond laser beam is needed to be focused into centrosymmetric crystal by a focused lens after passing through the polarizer 1. The generated third harmonic wave is collected by a detector. The polarizer 2 used here prevents the femtosecond laser beam from affecting the measured result. The crystal is mounted on a rotary stage. The angle $\theta$ changes as the target rotates. In a THG process, the incident fundamental wave (FW) and the transmitted THW generally contain a certain bandwidth in the frequency domain, which can be seen in Figure 3a. For a mode-locked femtosecond laser, the FW has $N$ equally spaced modes in the frequency domain. It is also the case for the THW. The spaces between the $i^{\text{th}}$ mode and the $i + 1$th mode are $\nu_{\text{rep}}$ and $3\nu_{\text{rep}}$ for FW and THW, respectively. The carrier-envelope offsets are $\nu_{\text{CEO}}$ and $3\nu_{\text{CEO}}$ for FW and THW, respectively, because the THG process triples the frequency of FW in the crystal. The $\lambda_{1,3}$ are used to denote the corresponding wavelengths of FW and THW, respectively, in Figure 3a. In this paper, we assume the focused beam has a typical Gaussian distribution in free space, and the characteristics are displayed in Figure 3b. Here, $w$ and $b$ are the beam waist and the confocal parameter of the Gaussian beam, respectively. The $Z_L$ and $Z_R$ are the coordinate value in the incident plane and the output plane, respectively. The crystal length $L = Z_R - Z_L$. The $d$ and $f$ are the diameter of the incident laser beam and the focal length of the focused lens, respectively.

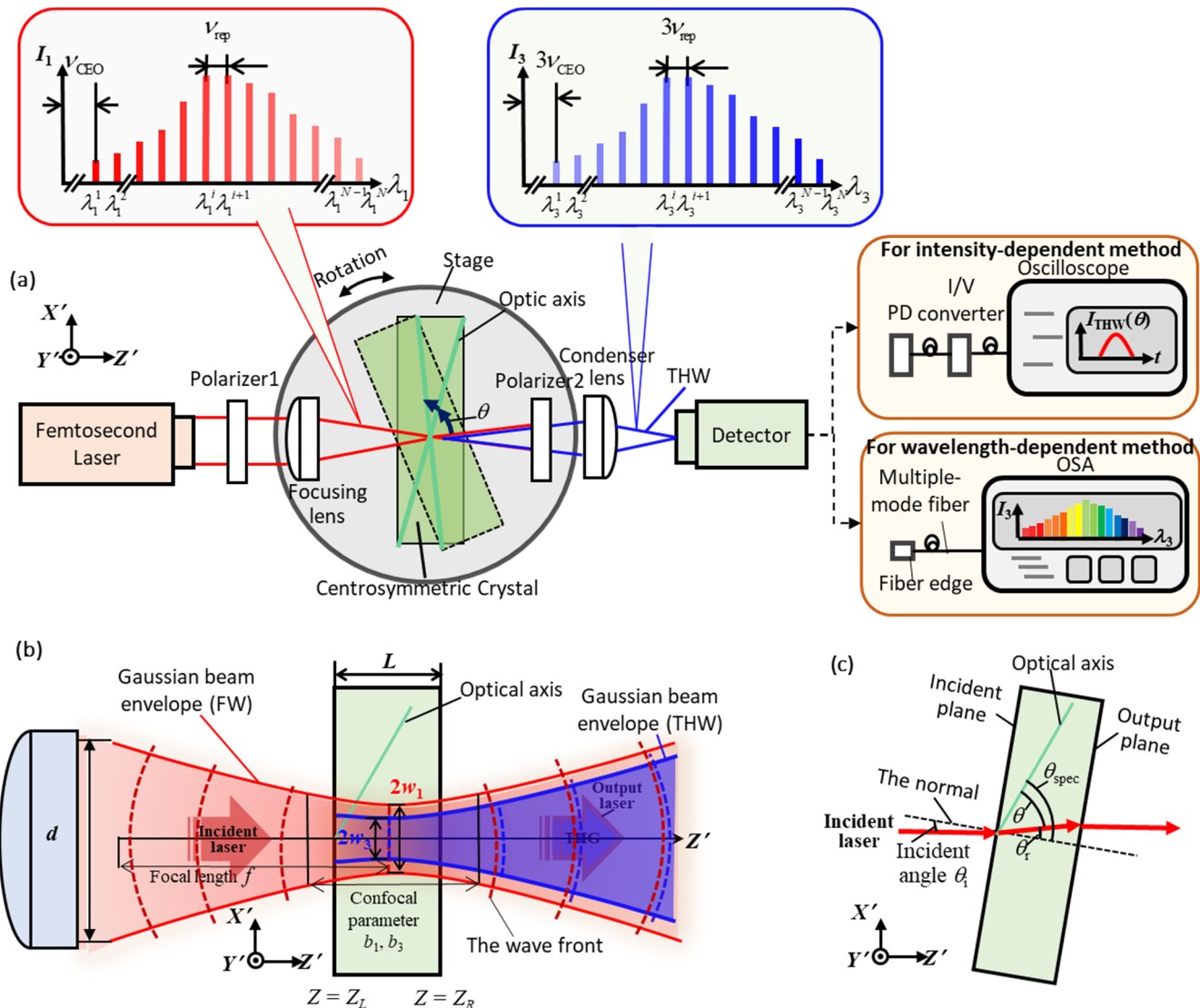

**Figure 3.** (**a**) The schematic of angle measurement based on direct THG in a centrosymmetric crystal; (**b**) The diagram of THG generated by focused beam; (**c**) The relationship between the incident angle $\theta_i$ and $\theta$.

The angle dependence of transmitted THG $I_3(\lambda_3, \theta)$ is given in Equation (4) [27], where c and $\varepsilon_0$ are the light speed and dielectric constant of free space, respectively. $I_1$ is the intensity of the fundamental wave. $\chi_{\text{eff}}^{(3)}$ is the third-order effective nonlinear coefficient, which is related to the phase-matching type and the $\chi^{(3)}$ matrix of crystal. Here, the $J_3$ function is crucial for the angle measurement. It is dependent on the phase-mismatching variation $\Delta k(\lambda_1, \theta)$. The equations for the calculation of $\Delta k$, $b_1$, and $w_1$ are given as Equation (6) to Equation (8), respectively. The beam waist and the confocal parameter for THW are $w_3 = \sqrt{3}w_1$ and $b_3 = b_1$, respectively [28]. $\chi_{\text{eff}}^{(3)}$ will be calculated in the next section corresponding to the investigated centrosymmetric crystals.

$$I_3(\lambda_3, \theta) = \frac{48\left(\chi_{\text{eff}}^{(3)}\right)^2 I_1^3}{n_3 n_1^3 \lambda_1^2 c^2 \varepsilon_0^2 w_1^4} |J_3(\Delta k(\lambda_1, \theta), b_1, Z_L, Z_R)|^2, 3\lambda_3 = \lambda_1 \tag{4}$$

$$|J_3|^2 = \left| \int_{z_L}^{z_R} \frac{e^{i\Delta k(\lambda_1, \theta)z} dz}{(1 + 2iz/b_1)^2} \right|^2 \tag{5}$$

$$\Delta k = \frac{6\pi}{\lambda_1}(n_{1e}(\lambda_1, \theta) - n_{3o}) \text{ Type I (Positive crystal)}$$
$$\Delta k = \frac{2\pi}{\lambda_1}(2n_{1o} + n_{1e}(\lambda_1, \theta) - 3n_{3e}(\lambda_3, \theta)) \text{ Type II (Negative crystal)}$$

(6)

$$b_1 = \frac{8f^2\lambda_1}{\pi d^2 n_1}$$

(7)

$$2w_1 = \frac{4f\lambda_1}{\pi d n_1}$$

(8)

Because the actual rotated angle for the measurement target is the incident angle $\theta_i$, the relationship between $\theta_i$ and $\theta$ should also be considered for the angle measurement. As shown in Figure 3c, $\theta_r$ is the refractive angle, and $\theta_{spec}$ is the angle between the optical axis and the normal direction, which is often around the phase-matching $\theta_m$. This enables the incident laser to easily achieve phase-matching conditions at the normal incidence. The relationship can be described by Snell's law as Equation (9), where $n_1$ can be calculated according to Equation (1), which is related to the dispersion and the angle $\theta$.

$$\sin\theta_i = n_1(\lambda_1, \theta)\sin\theta_r = n_1(\lambda_1, \theta)\sin(\theta_{spec} - \theta)$$

(9)

As reported in our previous studies, there are two types of angle measurement methods based on the phase-matching principle of THG: the intensity-dependent method and the wavelength-dependent method. For the former, a single photodiode (PD) can be used as the detector (see Figure 3a), which converts the angle-related intensity change into an electrical current output. Then, the electrical current is amplified to a voltage by an I/V converter, and the voltage is displayed on an oscilloscope. The dispersion of angle dependence is very important because a single PD can only detect the total intensity $I_{THW}$ of THW, which has a band from $\lambda_3^1$ to $\lambda_3^N$. Therefore, the characteristics in the frequency domain cannot be measured. The angle dependence of measured total intensity $I_{THW}(\theta)$ can be evaluated by Equation (10) [12].

$$I_{THW}(\theta) = \sum_{i=1}^{N} I_3(\lambda_3^i, \theta)$$

(10)

Figure 4 is used to visually explain the effect of dispersion. Supposing the angle dependence $I_3(\theta)$ is dispersion-less in the wavelength band as shown in Figure 4a, there is a less overlap area of angle dependence between the modes of $\lambda_3^1$, $\lambda_3^i$, $\lambda_3^{i+1}$, and $\lambda_3^N$. That means there is also less possibility for the angle sensitivity to be reduced between the modes. A relatively stronger total intensity $I_{THW}(\theta)$ can then be expected because the summation of Equation (10) will be large. Conversely, as shown in Figure 4b, if the angle dependence is dispersive, the expected angle measurement sensitivity will be weakened because of the large overlap area of angle dependence.

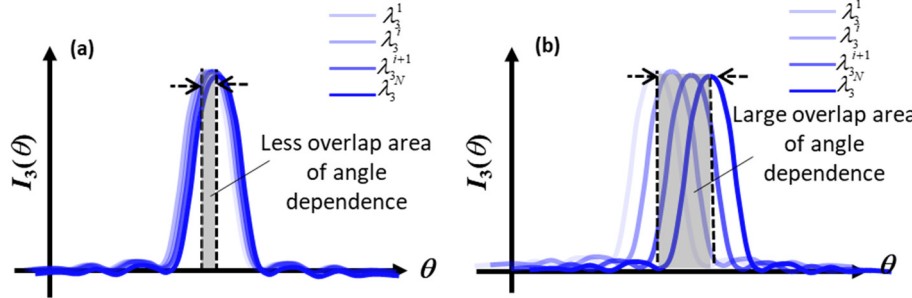

**Figure 4.** (**a**) The diagram of dispersion-less angle dependence $I_3(\theta)$; (**b**) The diagram of dispersive angle dependence $I_3(\theta)$.

Different from the intensity-dependent method, the wavelength-dependent method is based on the angle dependence dispersion. An optical spectrum analyzer (OSA) is used for THG detection in the frequency domain. As shown in Figure 5a, the different blue lines correspond to the angle dependence curve of different THG wavelengths $\lambda_3^1, \lambda_3^i, \lambda_3^{i+1}$, and $\lambda_3^N$. Due to the dispersion, the angle dependence curve for each wavelength is different. The corresponding phase-matching angles $\theta_m(\lambda_3)$ also change in the THG band. As the angle $\theta$ changes from $\theta_m(\lambda_3^1)$ to $\theta_m(\lambda_3^N)$, the corresponding wavelength that has the highest intensity should also change from $\lambda_3^1$ to $\lambda_3^N$. Therefore, an obvious spectrum peak shift can be seen in the frequency domain as illustrated in Figure 5b. To quantitatively characterize the angle-dependent peak shift, we use the weighted wavelength $\lambda_w(\theta_i)$ to denote peak wavelength [13], as given in Equation (11).

$$\lambda_w(\theta_i) = \frac{\sum\limits_{i=1}^{N} \lambda_3^i I_3(\lambda_3^i, \theta_i)}{\sum\limits_{i=1}^{N} I_3(\lambda_3^i, \theta_i)} \tag{11}$$

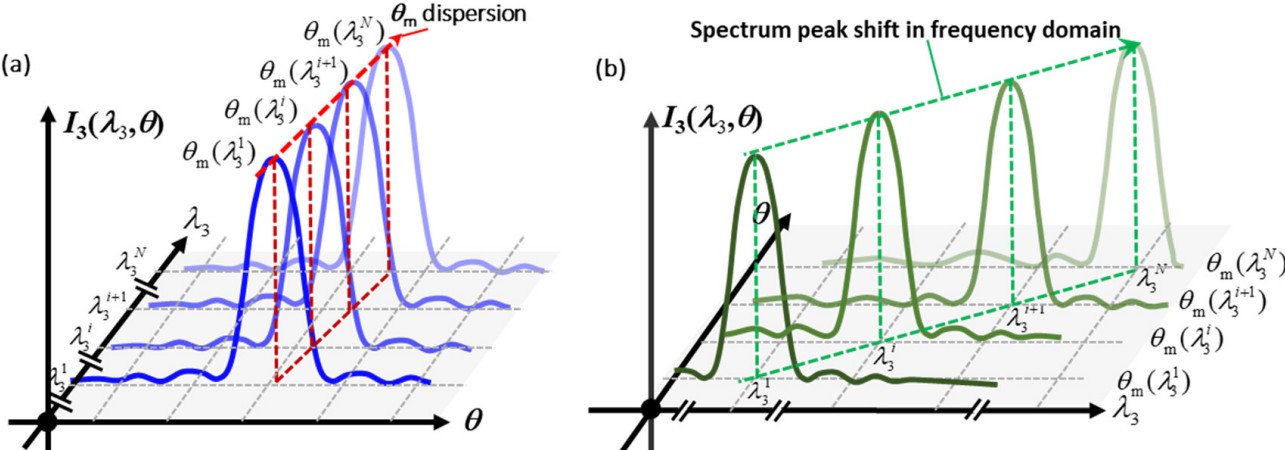

**Figure 5.** (**a**) The diagram of phase-matching angle $\theta_m$ dispersion; (**b**) The diagram of spectrum peak shift in the frequency domain.

## 3. Simulation Results and Discussion

### 3.1. The Investigated Centrosymmetric Crystals

In this paper, three centrosymmetric crystals, Alpha Barium Borate ($\alpha$-BBO), calcite, and rutile, are investigated considering their transparency, THG conversion efficiency, broad phase-matching range, and availability. Here, it should be clarified that $\alpha$-BBO is a typical centrosymmetric crystal, which is different from the non-centrosymmetric Beta Barium Borate ($\beta$-BBO) in our previous study, although they have a similar refractive index [26]. $\alpha$-BBO and calcite are negative crystals ($N_e < N_o$), and rutile is a positive crystal ($N_e > N_o$). The dispersion of the refractive index can be empirically calculated by the Sellmeier relation as shown in Equation (12). The coefficients are given in Table 1 from the literature. The third-order effective nonlinear coefficient $\chi_{eff}^{(3)}$ should be considered with the phase-matching type and the formation of $\chi^{(3)}$ matrix, which is dependent on the crystal class. Here, we discuss the type II phase-matching for $\alpha$-BBO and calcite of crystal class $\bar{3}$m, and the type I phase matching for rutile of crystal class 4/mmm, because of the high THG

conversion efficiency [26,29,30]. The calculation of $\chi_{\text{eff}}^{(3)}$ is given in Equation (12), and the parameters $\chi_{\text{eff}}^{(3)}(\theta_{\text{m}})$ can be found in the literature [30].

$$
\begin{aligned}
N_{\text{e}}(\lambda), N_{\text{o}}(\lambda) &= \sqrt{A + \frac{B}{\lambda^2 - C} - D\lambda^2} \quad \alpha-\text{BBO} \\
N_{\text{e}}(\lambda), N_{\text{o}}(\lambda) &= \sqrt{A + \frac{B\lambda^2}{\lambda^2 - C} - \frac{D\lambda^2}{\lambda^2 - E}} \quad \text{calcite} \\
N_{\text{e}}(\lambda), N_{\text{o}}(\lambda) &= \sqrt{A + \frac{B\lambda^2}{\lambda^2 - C}} \quad \text{rutile}
\end{aligned}
\tag{12}
$$

$$
\begin{aligned}
\chi_{\text{eff}}^{(3)} &= \frac{1}{3}\chi_{11}^{(3)}\cos^2\theta_{\text{m}} + \chi_{16}^{(3)}\sin^2\theta_{\text{m}} + \chi_{10}^{(3)}\sin 2\theta_{\text{m}}\sin 3\varphi \quad \alpha-\text{BBO, calcite} \\
\chi_{\text{eff}}^{(3)} &= \frac{1}{2}\cos^2\theta_{\text{m}}[\chi_{11}^{(3)}\sin^2 2\varphi + \chi_{18}^{(3)}(3\cos^2 2\varphi - 1)] + \chi_{16}^{(3)}\sin^2\theta_{\text{m}} \quad \text{rutile}
\end{aligned}
\tag{13}
$$

**Table 1.** The coefficients of Sellmeier Equations [26,31,32].

| Nonlinear Crystal | A | B | C | D | E |
|---|---|---|---|---|---|
| α-BBO | | | | | |
| $N_{\text{o}}$ | 2.7471 | 0.01878 $\mu m^2$ | 0.01822 $\mu m^2$ | 0.01354 $\mu m^{-2}$ | |
| $N_{\text{e}}$ | 2.3715 | 0.01224 $\mu m^2$ | 0.01667 $\mu m^2$ | 0.01516 $\mu m^{-2}$ | |
| Calcite | | | | | |
| $N_{\text{o}}$ | 1.7335 | 0.96464 $\mu m^{-2}$ | 1.94325 $\mu m^2$ | 1.8283 $\mu m^{-2}$ | 120 $\mu m^2$ |
| $N_{\text{e}}$ | 1.3585 | 0.82427 $\mu m^{-2}$ | 1.06689 $\mu m^2$ | 0.1442 $\mu m^{-2}$ | 120 $\mu m^2$ |
| Rutile | | | | | |
| $N_{\text{o}}$ | 5.913 | $2.441 \times 10^5$ $\mu m^{-2}$ | $0.803 \times 10^5$ $\mu m^2$ | | |
| $N_{\text{e}}$ | 7.197 | $3.322 \times 10^5$ $\mu m^{-2}$ | $0.843 \times 10^5$ $\mu m^2$ | | |

### 3.2. The Sensitivity Investigation of Angle Measurement Based on Direct THG

As illustrated before, the intensity-dependent and wavelength-dependent methods are based on the dispersion-less and dispersive properties of angle dependence. Here, we use the phase-matching angle $\theta_{\text{m}}(\lambda_1)$ to denote the dispersion speed in a certain wavelength range because $\theta_{\text{m}}$ represents the angle of the highest point in the angle dependence curve. We first consider a wavelength range from 900 nm to 2500 nm, which covers a commonly used range of femtosecond laser, to find a suitable wavelength range for intensity-dependent angle measurement. The simulation results are given in Figure 6a, where the lines correspond to α-BBO, calcite, and rutile. In the investigated wavelength range, the $\theta_{\text{m}}(\lambda_1)$ of rutile is monotonically decreasing from 79.8° to 34.2°, which indicates the angle dependence of rutile has a very significant dispersion in this range. Therefore, rutile is not an ideal crystal for intensity-dependent measurement in the investigated range. Compared with rutile, both the $\theta_{\text{m}}(\lambda_1)$ curves of α-BBO and calcite go down firstly from 900 nm to around 1700 nm, then keep stable from 1700 nm to 2000 nm, and finally slowly rise from 1700 nm to 2000 nm. We can see that the trends of α-BBO and calcite are very similar, and the range 1700 nm to 2000 nm is the ideal range for intensity-dependent angle measurement because of the low dispersion. In the dispersion-less area, the $\theta_{\text{m}}(\lambda_1)$ of α-BBO and calcite are around 33.0° and 27.1°. It is easy to understand that the derivative of $\theta_{\text{m}}(\lambda_1)$ in the dispersion-less area should be close to zero. Figure 6b shows the calculated phase-matching angle derivative in a unit of deg/nm. In this figure, the α-BBO and calcite correspond to the left vertical axis. It can be seen that the dispersion speed of $\theta_{\text{m}}(\lambda_1)$ is in the order of $10^{-3}$ deg/nm. Here, the red circles represent the positions having zero derivatives, which are 1803 nm and 1858 nm for calcite and α-BBO, respectively. As a

comparison, the derivative result of rutile is also given in the figure. It corresponds to the right vertical axis. We see the dispersion speed of $\theta_m(\lambda_1)$ is in the order of $10^{-2}$ deg/nm, which is much higher than the dispersion-less cases.

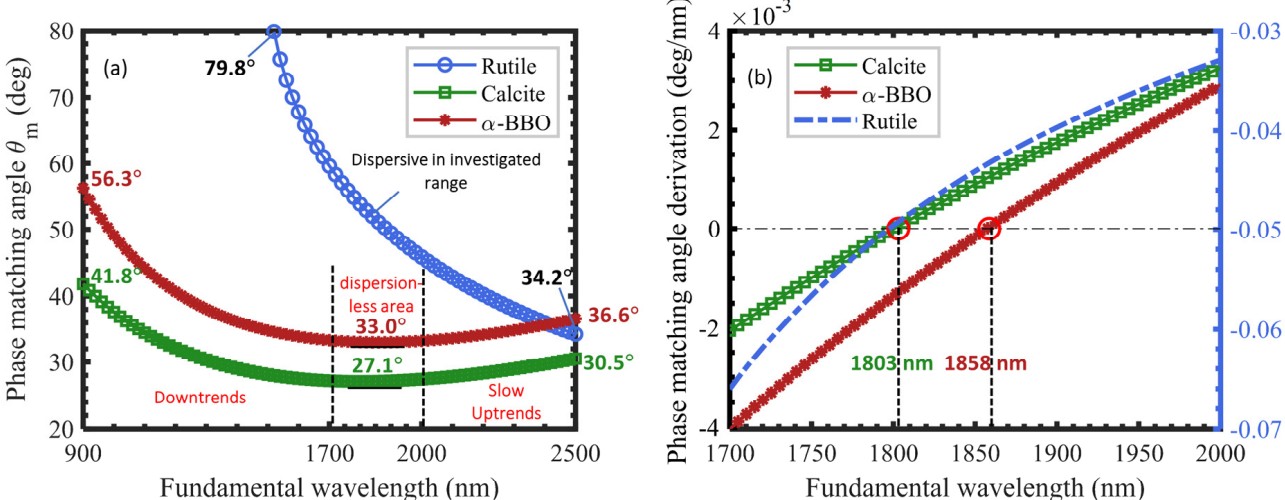

**Figure 6.** (**a**) The calculated results of phase-matching angle $\theta_m$ dispersion from 900 nm to 2500 nm; (**b**) The phase-matching angle derivative in the dispersion-less area from 1700 nm to 2000 nm.

To evaluate the angle measurement sensitivity of the intensity-dependent method, we calculate the angle dependence of $I_{THW}(\theta_i)$. In the simulation process, the parameters are set as: the focal length $f = 70$ mm, the Z positions of the crystal are $Z_L = -1$ mm and $Z_R = 1$ mm, respectively, the crystal length $L = Z_R - Z_L = 2$ mm, the diameter of incident femtosecond beam $d = 3.6$ mm. We assume the focused point (beam waist) is perfectly in the center of the crystal, and the azimuthal angle $\varphi = 30°$. The wavelength range of the incident femtosecond laser is set from 1800 nm to 1900 nm, covering the dispersion-less area. The space between the modes is 1 nm. Figure 7 shows the simulated results, where the index (a), (b), and (c) of subplots correspond to $\alpha$-BBO, calcite, and rutile, respectively. In each subplot, the lines in different colors correspond to the angle dependence $I_3(\theta_i)$ of different modes or the total intensity $I_{THW}(\theta_i)$. The horizontal axis denotes the angular displacement of the incident angle $\theta_i$. The left and right vertical axes denote the intensity of $I_3$ and $I_{THW}$, respectively. It is clear that $I_3(\theta_i)$ does not have an obvious change between the wavelength of 1800 nm, 1850 nm, and 1900 nm for $\alpha$-BBO and calcite, and the angle dependence of $I_{THW}(\theta_i)$ keeps the same shape as $I_3(\theta_i)$. The maximum total intensities of $I_{THW}$ are strengthened to 96.8 a.u. ($\alpha$-BBO), 93.0 a.u. (calcite), respectively. Conversely, due to the dispersion, the rutile has a much lower maximum total intensity $I_{THW}$ of 2.7 a.u. The measurement sensitivity is calculated from about 80% to 20% of $I_{THW}$, and they are $-0.087$ a.u./arcsec, $-0.108$ a.u./arcsec, 0.002 a.u./arcsec, for $\alpha$-BBO, calcite, and rutile, respectively shown in Figure 7.

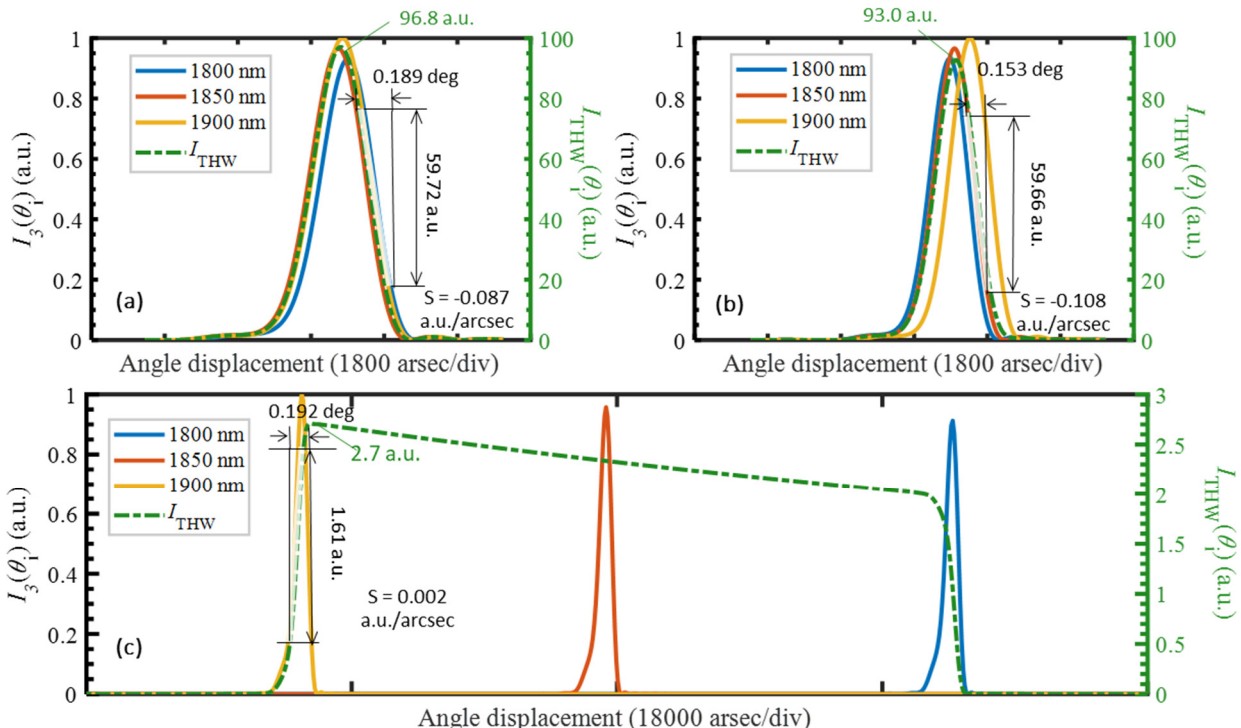

**Figure 7.** The calculated angle dependence of $I_3(\theta_i)$ and $I_{THW}$ ($\theta_i$) of (**a**) $\alpha$-BBO, (**b**) Calcite, and (**c**) Rutile.

For the wavelength-dependent method, the spectrum peak shift can be found except for the dispersion-less area (1700–2000 nm) of $\alpha$-BBO and calcite. In our previous work, a commercial femtosecond laser source was used for producing harmonic waves, and the central wavelength of the laser was around 1560 nm. Therefore, the wavelength-dependent method will be discussed near 1560 nm for future experimental work. In the simulation results, the parameters are the same as those of the intensity-dependent method. Figure 8 shows the calculated results for three crystals. In this figure, the letters a, b, and c correspond to the results of materials of $\alpha$-BBO, calcite, and rutile, respectively. The numbers 1 and 2 correspond to the 3-D plot and 2-D plot third harmonic intensity $I_3$, respectively. For each subplot, the color bar is used to indicate the amount of intensity. Here, we fixed the range from $-0.8°$ to $0.6°$ to compare the peak shifts. The results in Figure 8($a_1,b_1,c_1$) show that the peak shifts exist for all the investigated materials, and all the peak shifts toward the longer wavelength in the frequency domain as $\theta_i$ decreases. The peak shifts of $\alpha$-BBO and calcite both cover a range from around 500 nm to 540 nm. In contrast, the peak shift of rutile is relatively slow because it even cannot cover the wavelength range from 519 nm to 521 nm. The essence of peak shift is the point movement, which makes $\Delta n(\lambda_1, \theta) = $ const. This movement can be described by a contour map of $\Delta n(\lambda_1, \theta)$. It has been analyzed in our previous work on SHG. Therefore, it is not repeated here because of the similarity. The weighted wavelengths $\lambda_w(\theta_i)$ are shown in Figure 8($a_2,b_2,c_2$) by the white dotted lines to quantitatively evaluate peak shifts. The sensitivity is calculated by a linear fitting function to get the slope of $\lambda_w(\theta_i)$. The calculated sensitivities are $-0.0054$ nm/arcsec, $-0.0063$ nm/arcsec, and $-0.0002$ a.u./arcsec for $\alpha$-BBO, calcite, and rutile, respectively. The results suggest that $\alpha$-BBO and calcite have better measurement performance in the wavelength around 1560 nm.

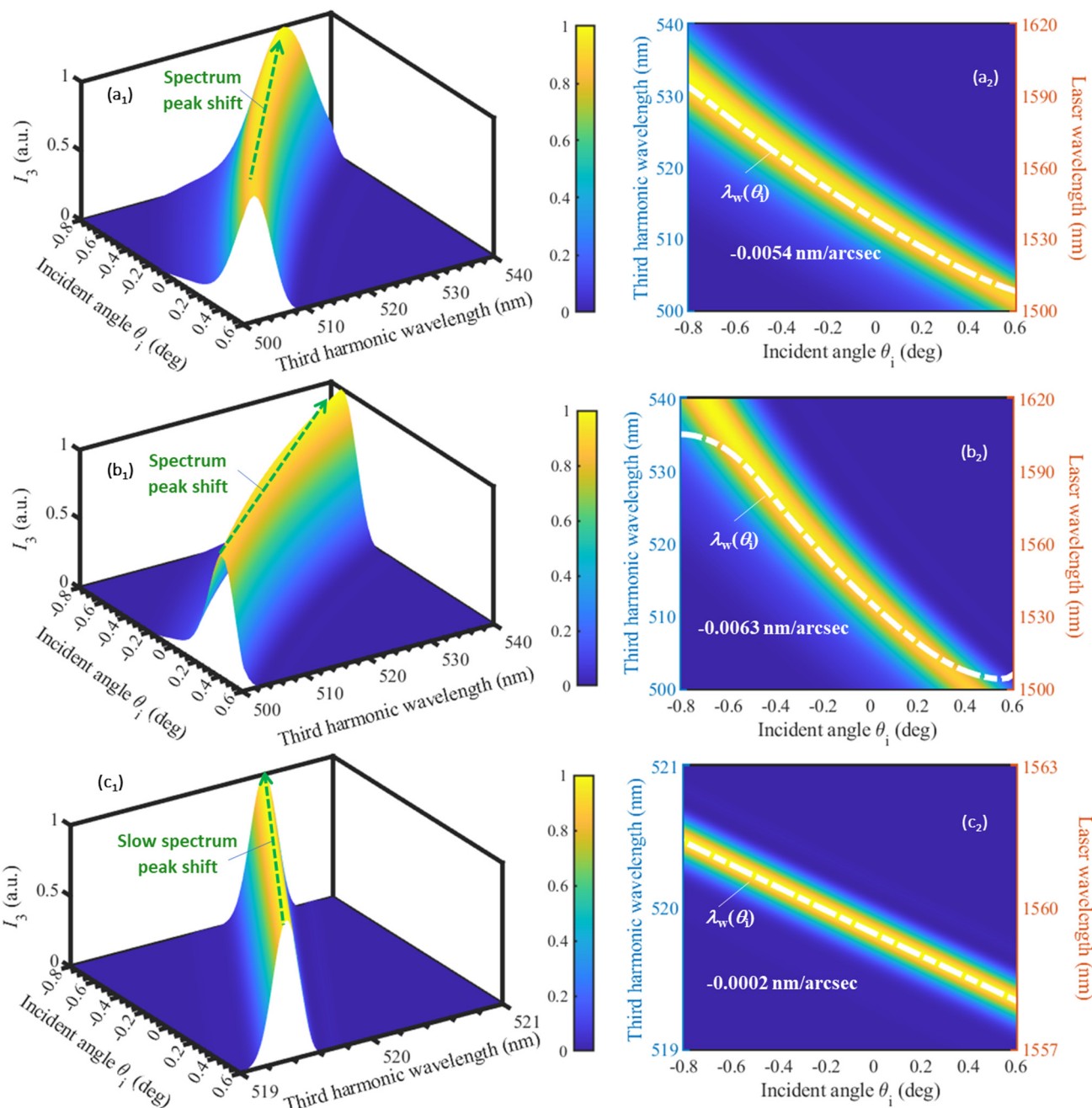

**Figure 8.** ($a_1$,$b_1$,$c_1$) The 3-D plot $I_3$; ($a_2$,$b_2$,$c_2$) The 2-D plot $I_3$. The letters a, b, and c correspond to the materials of $\alpha$-BBO, calcite, and rutile in order. The green dotted arrows show the spectrum peak shifts in frequency domain. The white dotted lines denote the weighted wavelength $\lambda_w(\theta_i)$.

### 3.3. The Focusing Parameter's Effects on Angle Measurement

The influence of the focusing parameters on the THG angle measurement is investigated by changing the focal length $f$ and the focused position. Here, we only put the results of calcite crystal because of its better sensitivity performance compared with rutile. The results are similar to those of $\alpha$-BBO. In the simulation, the crystal length $L = 2$ mm, $d = 3.6$ mm, $\lambda_1 = 1850$ nm. The focused point is in the center of the crystal and the azimuthal angle $\varphi = 30°$. Figure 9 shows the results of angle dependence $I_3(\theta_i)$, where the lines in different colors correspond to different focal lengths. In Figure 9a, the range of incident angle $\theta_i$ is from $-13°$ to $1°$. The peak intensities corresponding to the focal lengths are noted in the figure. The peak intensities are not monotonically decreasing as the focal length $f$

increases, and the highest intensity is 1.00 a.u. appear as $f$ = 70 mm. The overall trend of the peak intensity is that it increases from 0.74 a.u. (10 mm) to 1.00 a.u. (70 mm) then decreases from 1.00 a.u. (70 mm) to 0.51 a.u. (150 mm). We can also see the width of the angle dependence curve is reduced as the focal length becomes longer from $f$ = 10 mm (11.9 deg.) to $f$ = 150 mm (0.5 deg.). In Figure 9b, $\theta_i$ only covers the range from $-1°$ to $0.5°$, and the sensitivity result is shown except for the $f$ = 10 mm because of the relatively low value. We can see the maximum sensitivity result is $-4.21$ a.u./deg when $f$ = 100 mm, rather than $f$ = 70 mm, which has the highest peak intensity. This is because the $f$ = 100 mm case has a similar high peak intensity (0.96 a.u.) but has a relatively narrower angle dependence range compared with the $f$ = 70 mm case. Therefore, better angle measurement can be expected from the setting of $f$ = 100 mm.

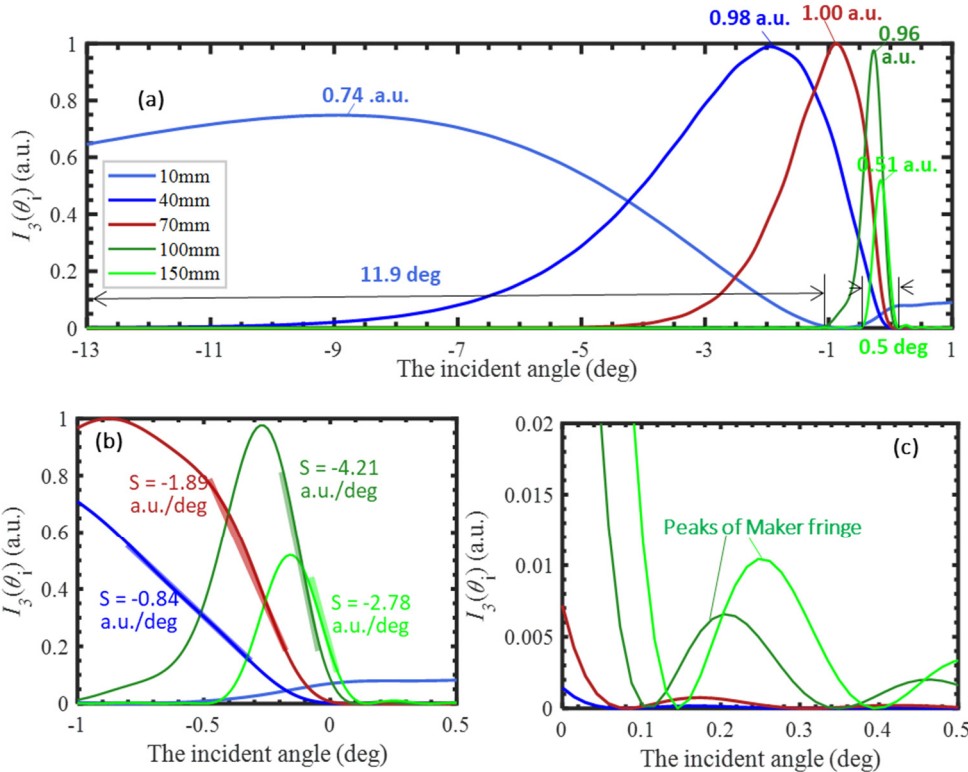

**Figure 9.** (**a**) The results of angle dependence $I_3(\theta_i)$ from $-13°$ to $1°$; (**b**) The angle dependence $I_3(\theta_i)$ from $-1°$ to $0.5°$; (**c**) The zoom in from (**b**), where the horizontal axis and vertical axis are limited in the range of $0°$–$0.5°$ and 0 a.u.–0.02 a.u., respectively.

We also give a qualitative explanation of the results of Figure 9a. It can be seen from Equation (4) to Equation (8) that the angle dependence $I_3(\theta_i)$ is related to both the beam waist $w_1$ and the confocal parameter $b_1$. As the focal length $f$ decreases, both the beam waist $w_1$ and the confocal parameter $b_1$ are reduced in Equations (7) and (8). On the one hand, as in Equation (4), the decrease of $w_1$ makes the $I_3$ increase directly. On the other hand, the decrease of confocal parameter $b_1$ will make the $|J_3|^2$ change because $J_3$ is related to the term $1/(1 + 2iz/b_1)$. After a Taylor expansion of $1/(1 + 2iz/b_1)$, $|J_3|^2$ has the following form as shown in Equation (14),

$$
\begin{aligned}
|J_3|^2 &= \left| J_3^{(1)} + J_3^{(2)} \right|^2 \\
J_3^{(1)} &= \int_{z_L}^{z_R} e^{i\Delta k(\theta)z} dz = -L \operatorname{sinc}(\Delta kL/2) \\
J_3^{(2)} &= \int_{z_L}^{z_R} \sum_{i=1}^{\infty} (-1)^i (i+1) \left(\frac{2i}{b}\right)^i z^i e^{i\Delta k(\theta)z} dz
\end{aligned}
\tag{14}
$$

where the decrease of $b_1$ will make the part of $J_3^{(2)}$ to dominate $J_3^{(1)}$. It will induce an extra phase in $|J_3|^2$, which is harmful to the phase-matching when $\Delta k(\theta_m) = 0$. Therefore, there should be a trade-off between the beam waist $w_1$ and confocal parameter $b_1$. The focal length of 100 mm is the best choice, which has a Rayleigh length of 2.2 mm. In addition, as can be found in Equation (14), a longer focal length $f$ makes the $J_3^{(2)}$ smaller and $|J_3|^2$ will degenerate to the $\mathrm{sinc}^2(\Delta k(\theta)L/2)$ function form. Then, the side peaks of $\mathrm{sinc}^2(\Delta k(\theta)L/2)$ in the Maker fringe area also can be seen in Figure 9c [33].

In the optimized case of $f = 100$ mm, the effect of the beam waist position is considered in Figure 10. Lines and circles are used to denote the beam waist positions of $\pm 0.2$ mm, $\pm 0.4$ mm, $\pm 0.6$ mm, $\pm 0.8$ mm, and $\pm 1$ mm relative to the crystal center. The maximum intensity is obtained in the center of the crystal at 1.00 a.u., which can be seen in the red line. This is because the central position makes the high-intensity area of the Gaussian beam to be fully used for the THG process. As the beam waist gradually moves away, the peak intensity falls from 0.95 a.u. ($\pm 0.2$ mm) to 0.51 a.u. ($\pm 1$ mm). The symmetric results for the positive/negative beam waist movements are the consequence of the Gaussian beam's symmetric shape. In addition, the angle dependence areas of $I_3$ are both close to 0.63 deg, as shown in the figure. Thus, the best measurement sensitivity should be obtained when the beam waist is in the center associated with the highest intensity.

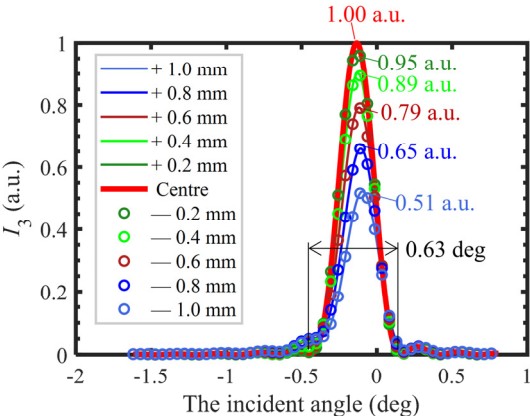

**Figure 10.** The figure for showing the effect of different focused positions on angle dependence. The lines/circles in different colors correspond to the positive/negative beam waist movements relative to the crystal center.

## 4. Conclusions

In this paper, we have proposed the intensity-dependent and wavelength-dependent angle measurement methods by THG in centrosymmetric crystals. Three prospective crystals, including the $\alpha$-BBO, calcite, and rutile, have been considered in this research for the two angle measurement methods. The effects of the focusing parameters on the angle dependence have also been considered. The optimized focusing parameters have been found for future experimental work.

For the intensity-dependent method, the type I phase-matching angle of rutile and the type II phase-matching angle of $\alpha$-BBO have been calculated in the wavelength range from 900 nm to 2500 nm. The results show that the type II phase-matching angle has a good performance due to its dispersion-less property in the wavelength range of 1700–2000 nm. The sensitivities of angle measurement have been evaluated for calcite and $\alpha$-BBO, which are $-0.108$ a.u./arcsec and $-0.087$ a.u./arcsec, respectively. They are much higher than the sensitivity of 0.002 a.u./arcsec for rutile. The sensitivity of the wavelength-dependent method has been evaluated in the dispersive range of around 1560 nm. The peak shift in the frequency domain has been shown for the three investigated crystals. The weighted wavelengths $\lambda_w$ have been used to characterize the peak shift speeds, which were

−0.0054 nm/arcsec, −0.0063 nm/arcsec, and −0.0002 a.u./arcsec for *α*-BBO, calcite, and rutile, respectively. *α*-BBO and calcite also have fast peak shifts around 1560 nm.

In addition, the focal length *f* has been changed from 10 mm to 150 mm for investigating the effect of *f* on the angle measurement. The optimized focal length has been obtained as *f* = 100 mm. It has been demonstrated that the maximum sensitivity is achieved when the focus position is in the center of the crystal.

**Author Contributions:** Methodology, K.L., R.S. and H.M.; Formal analysis, K.L.; Investigation, J.L., R.S. and H.S.; Data curation, J.L. and Z.Z.; Writing—original draft, K.L.; Writing—review & editing, H.M.; Supervision, W.G. All authors have read and agreed to the published version of the manuscript.

**Funding:** This work is supported by the Japan Society for the Promotion of Science (JSPS) 20H00211.

**Institutional Review Board Statement:** Not applicable.

**Informed Consent Statement:** Not applicable.

**Data Availability Statement:** The data presented in this study are available on request from the corresponding author.

**Acknowledgments:** Li Kuangyi would like to thank the Chinese Scholarship Council (CSC) for living cost support.

**Conflicts of Interest:** The authors declare no conflict of interest. The funders had no role in the design of the study; in the collection, analyses, or interpretation of data; in the writing of the manuscript, or in the decision to publish the results.

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
