# Peer review of "Investigation of Angle Measurement Based on Direct Third Harmonic Generation in Centrosymmetric Crystals"

_applsci, doi:10.3390/app13020996_

Round 1

Reviewer 1 Report

Dear Author,

This work is highly intriguing, as well as having strong objectivity, originality, and a scientifically sound preparation.

As a result of reading your manuscript, I have a few general points to make. If you make the necessary edits and submit the revised version again, your manuscript will consider for publication.

Reviewer 2 Report

The present manuscript entitled Investigation of angle measurement based on direct third har- 2 monic generation in centrosymmetric crystal. The manuscript is well written and the results are interpreted properly with suitable experiments. However this manuscript can be recommended for publication in the Applied Sciences after the minor revision by addressing the comments given below,

1. Add some more information about novelty in introduction part

2. Most of the reference are very old; hence may cite recent reference related to this work

3. The authors are requested to check the grammatical and typo errors before submission.
